# Label-Free Quantification Mass Spectrometry Identifies Protein Markers of Chemotherapy Response in High-Grade Serous Ovarian Cancer

**DOI:** 10.3390/cancers15072172

**Published:** 2023-04-06

**Authors:** Georgia Arentz, Parul Mittal, Manuela Klingler-Hoffmann, Mark R. Condina, Carmela Ricciardelli, Noor A. Lokman, Gurjeet Kaur, Martin K. Oehler, Peter Hoffmann

**Affiliations:** 1Adelaide Proteomics Centre, School of Biological Sciences, The University of Adelaide, Adelaide, SA 5005, Australia; 2Clinical & Health Sciences, University of South Australia, Adelaide, SA 5001, Australia; 3Future Industries Institute, Mawson Lakes Campus, University of South Australia, Mawson Lakes, SA 5095, Australia; 4Discipline of Obstetrics and Gynecology, Adelaide Medical School, Robinson Research Institute, University of Adelaide, Adelaide, SA 5000, Australia; 5Institute for Research in Molecular Medicine, University Sains Malaysia, Minden 11800, Pulau Pinang, Malaysia; 6Department of Gynecological Oncology, Royal Adelaide Hospital, Adelaide, SA 5005, Australia

**Keywords:** chemotherapy resistance, ovarian cancer, mass spectrometry

## Abstract

**Simple Summary:**

Ovarian cancer is the most lethal gynecological cancer, despite advances in patient stratification and treatment. Despite initial response to first-line treatment, most patients suffer a relapse and die from treatment-resistant disease. We identified three predictive protein biomarkers for chemotherapy response using primary ovarian cancer samples and showed their concurrent regulation in a chemoresistant cell line. Those markers will potentially help in understanding the mechanisms of chemotherapy resistance.

**Abstract:**

Eighty percent of ovarian cancer patients initially respond to chemotherapy, but the majority eventually experience a relapse and die from the disease with acquired chemoresistance. In addition, 20% of patients do not respond to treatment at all, as their disease is intrinsically chemotherapy resistant. Data-independent acquisition nano-flow liquid chromatography–mass spectrometry (DIA LC-MS) identified the three protein markers: gelsolin (GSN), calmodulin (CALM1), and thioredoxin (TXN), to be elevated in high-grade serous ovarian cancer (HGSOC) tissues from patients that responded to chemotherapy compared to those who did not; the differential expression of the three protein markers was confirmed by immunohistochemistry. Analysis of the online GENT2 database showed that mRNA levels of GSN, CALM1, and TXN were decreased in HGSOC compared to fallopian tube epithelium. Elevated levels of GSN and TXN mRNA expression correlated with increased overall and progression-free survival, respectively, in a Kaplan–Meier analysis of a large online repository of HGSOC patient data. Importantly, differential expression of the three protein markers was further confirmed when comparing parental OVCAR-5 cells to carboplatin-resistant OVCAR-5 cells using DIA LC-MS analysis. Our findings suggest that GSN, CALM1, and TXN may be useful biomarkers for predicting chemotherapy response and understanding the mechanisms of chemotherapy resistance. Proteomic data are available via ProteomeXchange with identifier PXD033785.

## 1. Introduction

Ovarian cancer is the eighth most common cancer in women and a major cause for cancer-related death with more than 313,000 cases diagnosed in 2020 worldwide [1]. It was estimated that there would be 1815 new ovarian cancer cases and 1016 deaths from this disease in 2022 in Australia [2]. There are two main reasons for this poor outcome. Firstly, there is currently no effective screening strategy in place to systematically detect ovarian cancer at an early stage when the tumor is still confined to the ovary. In more than 70% of cases, patients are diagnosed when the cancer has already advanced to stage III or IV. Secondly, although 80% of patients initially respond to first line carboplatin + paclitaxel chemotherapy, most will relapse. In addition, 20–40% of patients do not respond at all to treatment, meaning their cancer has some form of innate resistance.

During the last 40 years, only minimal progress has been made and ovarian cancer remains the most lethal gynecological cancer. A reliable biomarker which could be used for population-wide screening to detect early disease does not exist. In addition, as the symptoms for ovarian cancer, such as persistent abdominal bloating, excessive fatigue or lethargy, or the need to urinate often or urgently, are subtle, the disease is therefore often described as silent [3]. The current diagnosis of ovarian cancer involves physical examination, a blood test for carbohydrate antigen 125 (CA 125) levels, and imaging such as a pelvic ultrasound or CT scan. However, studies have shown the low sensitivity and specificity of serum CA125 assay in detecting ovarian cases in initial stages, reported to be high in only 23 to 50% of stage I ovarian cancer cases [4].

By the time ovarian cancer is diagnosed, the cancer has often spread to other organs and over 70% of patients present with advanced metastatic disease and require a combination of tumor debulking surgery and chemotherapy. Although considerable progress has been made in understanding the genetic diversity of ovarian cancer that correlates with patient outcome, this has not translated into personalized management [5]. Standard treatment regimens for ovarian cancer usually involve a combination of carboplatin and paclitaxel [6]. Following treatment, 80% of women initially respond to chemotherapy, but the majority eventually relapse with acquired carboplatin- and paclitaxel-resistant disease [7,8], and are treated with other chemotherapeutic drugs such as liposomal doxorubicin or gemcitabine [9,10,11]. Furthermore, 20% of patients who receive systemic treatment do not respond at all, meaning that their cancer has some form of innate resistance. Preclinical investigations imply that in vitro drug response assays, which test patient tumor tissues for chemosensitivity/chemoresistance to specific compounds, could improve treatment efficacy [12]. However, due to the practical difficulties associated with collecting, maintaining, and treating primary cell cultures, in vitro drug response assays are not yet part of clinical practice.

How cancer cells evade or evolve to escape chemotherapy is yet to be fully understood. Some mechanisms include upregulation of ion pumps/channels, drug influx and efflux pathways, epithelial–mesenchymal transition, epigenetic mechanisms, and DNA damage-repair machinery [13,14]. Guo et al. summarized the molecular mechanism of platinum resistance in ovarian cancer induced by miRNA, long non-coding RNA, circular RNA, and epithelial–mesenchymal transition [15]. Moreover, various biomarkers of platinum resistance in ovarian cancer have been reported including BRCA1/2, p53, CA125, DNA methylation, and miRNAs [16]. However, none of these markers have been identified in our proteomic study indicating that they are of low abundance. Moreover, the sensitivity and specificity of these biomarkers needs to be validated prior to clinical use. Additionally, further studies are required to identify the specific molecular markers for chemotherapy response in high-grade serous ovarian cancer (HGSOC) patients. Recently we have generated carboplatin-resistant cancer cell lines (OVCAR-5 and CaOV3). Label-free proteomics analysis on these two pairs of parental and carboplatin-resistant cell lines showed a shared dysregulation of cytokine and type 1 interferon signaling, potentially revealing a common molecular feature of chemoresistance [17]. Here, we aimed to identify protein markers of chemotherapy response in HGSOC patients who did not respond to platinum-based chemotherapy. 

The International Federation of Obstetrics and Gynecology (FIGO) stage IIIC serous ovarian tissues that had been formalin-fixed and paraffin-embedded (FFPE) were retrieved from pathology archives and used in this study. As the tissues were collected prior to chemotherapy, the differentially expressed proteins could indicate innate chemoresistance. Using data-independent acquisition (DIA) nano-flow liquid chromatography mass spectrometry (LC-MS) on FFPE tissue samples, the expression of three proteins—gelsolin (GSN), calmodulin (CALM1), and thioredoxin (TXN) —was found to be significantly increased in patients who responded to chemotherapy versus those who did not. The expression of these proteins was further compared in a larger sample cohort by immunohistochemistry (IHC). Analysis of the online GENT2 database found that GSN, CALM1, and TXN mRNA expression was reduced in HGSOC compared to fallopian tube (FT) epithelium, the site of origin of the majority of HGSOC [18]. Furthermore, Kaplan–Meier analysis revealed elevated levels of GSN and TXN mRNA expression correlated with increased overall survival (OS) and progression-free survival (PFS), respectively. The relative expression of the proteins was further confirmed by DIA LC–MS in the ovarian cancer cell line OVCAR-5, where parental cells were compared to cells that had been treated with carboplatin to develop acquired chemoresistance [17]. The differential expression of GSN, CALM1, and TXN in the parental OVCAR-5 cells compared to carboplatin-resistant (CBPR) OVCAR-5 cells showed similar expression, as observed in the patient cohort. These results indicate that GSN, CALM1, and TXN may be useful biomarkers to assess HGSOC progression and chemotherapy response, warranting further investigations. 

## 2. Materials and Methods

### 2.1. Sample Collection and TISSUE Specimens

FIGO stage IIIC HGSOC tissues (n = 12) were obtained from patients undergoing surgery at the Department of Gynecological Oncology (Royal Adelaide Hospital, Adelaide, Australia) with approval by the hospital ethics committee and written informed consent (protocol number: 140101). Tissues were processed by the clinical pathology laboratory according to their standard procedure, including FFPE. Patients were classified as platinum-sensitive if they did not progress within 6 months of completing the chemotherapy treatment. Patients were classified as non-response/incomplete response if they did not respond to chemotherapy treatment or relapsed within 6 months. The clinicopathological patient information is provided in Table 1.

Two tissue microarrays (TMAs) were constructed containing 31 patients, including 21 who responded to chemotherapy, and 8 who had not. A further 2 patients had unknown chemotherapy response. Detailed patient information is listed in Appendix A and a the construction of the TMA has been described by us previously [19].

### 2.2. Laser Microdissection and Protein Extraction of the Ovarian Cancer Tissues

FFPE tissue blocks (n = 6 chemo-responsive and n = 6 non-responsive tissues) were water-bath-mounted onto PEN membrane slides (MicroDissect, Herborn, Germany) at 4 µm, as previously described [20,21]. Briefly, the slides were deparaffinized by heating at 60 °C for 1 h, followed by rehydration using xylene (Chem-Supply, Gillman, Australia) for 90 s and then rinsed for 60 s with 100% ethanol (Merck, Bayswater, Australia). Slides were hematoxylin-stained and scanned using a NanoZoomer (Hamamatsu, Japan) at 43 × resolution (0.23 µm/pixel), and the tissues were annotated by an experienced pathologist using NPD.view 2.6.13 (Hamamatsu, Beijing, China). Approximately 1 mm^2^ per region of each tumor tissue was excised by laser capture microdissection (LCM) using a Leica microscope (Leica Microsystems, Wetzlar, Germany) into individual centrifuge tubes. Excised tissues were lysed using 2% SDS (GE Healthcare, Parramata, Australia) in 10 mM citric acid, pH 6 (citric acid monohydrate, Sigma-Aldrich, Japan), followed by antigen retrieval at 98 °C for 90 min. The total protein concentration of each sample was estimated using a NanoDrop 2000 (Thermo-Fisher Scientific, Waltham, MA, USA) at 280 nm.

### 2.3. Generation of Carboplatin-Resistant (CBPR) OVCAR-5 Cells

OVCAR-5, the human ovarian cancer cell line, was obtained from Dr. Thomas Hamilton (Fox Chase Cancer Centre, Philadelphia, PA) and cultured under similar conditions to those previously described by us [17,22]. Briefly, the carboplatin-resistant (CBPR) OVCAR-5 cells were obtained by treating parental OVCAR-5 cells with 25 µM of carboplatin for 24 h. (Hospira Australia Pty Ltd., Mulgrave, Australia) followed by 72 h of recovery phase. This step was repeated for 8 cycles, followed by the development phase for 8–10 weeks. The 25 µM of carboplatin was chosen because of the half maximal inhibitory concentration (IC50) of carboplatin, which was calculated using 3 independent experiments performed in triplicate. The cell survival was calculated using MTT assay (Sigma-Aldrich, Castle Hill, Australia), as per the manufacturer’s recommendation [22].Protein extraction was performed using 2% (w/v) SDS lysis buffer with protease inhibitor cocktail (Roche, Basel, Switzerland). The DNA was sheared on ice using Bioruptor (Diagenode, Liege, Belgium) for 6 × 30 s cycles with one min interval in between each cycle. Protein concentration was quantified using an EZQTM protein assay (Thermo Fisher Scientific, USA), as per the manufacturer’s protocol.

### 2.4. Protein Digestion

Protein extracts from Section 2.2 and Section 2.3 were digested using filter-aided sample preparation (FASP) protocol with minor modifications [21,23]. Briefly, samples were denatured by adding 200 µL of 7M urea in 100 mM ammonium bicarbonate (Merck, Bayswater, Australia), followed by reduction at 20 °C for 1 h with dithiothreitol (Sigma-Aldrich, Castle Hill, Australia) to a final concentration of 50 mM. To remove the traces of glycerin, Sartorius Vivacon 500 ultrafiltration spin columns (10,000 MWCO HY) were pre-rinsed with 100 mM of ammonium bicarbonate. Samples were then loaded into the Vivacon spin columns and centrifuged for 10 min at 14,000 *g*. Thereafter, alkylation was performed by adding 100 μL of 55 mM iodoacetamide (IAA, GE Healthcare, Danderyd, Sweden) in 25 mM ammonium bicarbonate in the dark at room temperature for 20 min. Samples were then centrifuged and washed twice with 100 μL of 50 mM ammonium bicarbonate. Trypsin digestion (trypsin gold, Promega, Madison, WI, USA) was performed at an enzyme to substrate ratio of 1:50 in 100 μL of 25 mM ammonium bicarbonate at 37 °C overnight.

### 2.5. Identification of Proteins by DDA Nano-LC-ESI-MS/MS

Nano-LC-ESI-MS/MS was performed using an Ultimate 3000 RSLC system (Thermo-Fisher Scientific, USA) coupled to an Impact HD™ QTOF mass spectrometer (Bruker Daltonics, Bremen, Germany) via an Advance CaptiveSpray source (Bruker Daltonics, Bremen, Germany). Approximately 100 ng of each patient sample and approximately 400 ng of the parental and CBPR OVCAR-5 cell digests were analyzed by data-dependent acquisition (DDA) nano-LC-ESI-MS/MS. This was done to generate protein target lists and spectral libraries for the DIA analysis. Peptide samples were pre-concentrated onto a C18 trapping column (Acclaim PepMap100 C18 75 μm × 20 mm, Thermo-Fisher Scientific, Waltham, MA, USA) at a flow rate of 5 μL/min in 2% (*v*/*v*) acetonitrile 0.1% (*v*/*v*) formic acid (FA) for 10 min. Peptides were separated at a flow rate of 200 nL/min using a 75 μm × 50 cm Acclaim PepMap100 Thermo-Fisher Scientific C18 column. The gradient used was linear from 5 to 45% buffer B over 190 min, followed by gradual increase to 90% buffer B for 30 s and then held at 90% B for 20 min, followed by equilibration with 5% A for 20 min. Buffer B was 80% acetonitrile in 0.1% FA (*v*/*v*) and buffer A was 5% acetonitrile in 0.1% FA (*v*/*v*). The data were acquired over the mass range of 300 to 2200 *m/z* using Bruker’s Shotgun Instant Expertise™ method. The details of the method used are described by us in detail [21]. The ionization source settings used were capillary voltage of 1300 V, end plate offset of 500 V, and drying gas at 3 L min^−1^ at 150 °C. 

### 2.6. DDA Nano-LC-ESI-MS/MS Data Analysis 

To generate spectral libraries for the DIA assay, the DDA spectra were analyzed using the MaxQuant software (version 1.5.2.8) with the Andromeda search engine [24] against the UniProt human database. The standard Bruker Q-TOF settings with a precursor mass-error tolerance of 40 ppm and variable oxidation as methionine, carbamidomethylation of cysteines as fixed modification, and trypsin as the digestion enzyme with up to two missed cleavage was used. The false discovery rate (FDR) was set to 1% for both proteins and peptides, with a minimum peptide length of 7 amino acids.

### 2.7. Quantification of Protein(s) of Interest by DIA Nano-LC-ESI-MS

Nano-LC was performed as described above, loading either 100 ng of each patient sample or 400 ng of the parental and CBPR OVCAR-5 cell digests. An Ultimate 3000 RSLC system was coupled to an Impact HD™ QTOF mass spectrometer and data were acquired using Bruker’s Middle Band CID™ method. This data-independent acquisition (DIA) method scans a mass range of *m/z* 375 to 1206 in 26 Da increments and CID is performed with increasing collision energies of 20 to 36. The acquired DIA data were analyzed in the Skyline software (Version 3.1.0.7382) [25]. Spectral libraries were generated in Skyline from the MaxQuant “msms’ files, produced from the DDA nano-LC-ESI-MS/MS experiments, and matched against a background proteome of human FASTA sequences downloaded from the UniProt. The spectra collected by DIA LC-MS was then matched back to all the peptides identified in the spectral libraries and relative quantification was performed. The Skyline peptide and transition settings were as follows: trypsin was specified as the cleavage enzyme with a maximum of 1 missed cleavage, precursor charge states 2 and 3, product ion charges 1 and 2, ion types y and b from ion 2 to ion 6, an ion match tolerance of 0.1 *m/z*, a MS/MS filtering DIA isolation scheme from *m/z* 375 to 1206 (26 Da windows), a retention time window of 5 min, and a resolution of 10,000. A minimum of 2 unique and only unmodified tryptic peptides were used in the analysis for each protein. The DIA data were analyzed in Skyline software, where the transition and the retention time for each peptide were checked manually. The quantification was performed using the summed area intensity of each peptide in each sample and the standard deviations and *p* values using un-paired *t*-tests were calculated using the GraphPad Prism 6 v008 (GraphPad Software, La Jolla, CA, USA).

### 2.8. Validation by Immunohistochemistry (IHC)

Two TMAs (TMA 1 and TMA 2) containing a total of 31 patients; 21 responders, 8 non-responders, and 2 outcome unknowns were analyzed by IHC, as previously described [26]. Briefly, 6 µm TMA sections were placed on plain glass slides, dewaxed, and rehydrated with xylene (5 min twice) and ethanol (2 min twice). The endogenous peroxidase activity was quenched with 1% hydrogen peroxidase in PBS, followed by antigen retrieval using 10 mM citric acid (pH 6), at 100 °C for 10 min in a microwave (Sixth Sense, Whirlpool, Dandenong South, Australia). TMA sections were then incubated overnight at 4 °C with the rabbit polyclonal antibody calmodulin CALM1 (diluted 1:200, proteintech), gelsolin GSN (diluted 1:100, proteintech), and thioredoxin TXN (diluted 1:100, proteintech) in 5% goat serum blocking buffer. For negative control, tissues were incubated with 5% goat serum without any antibody. The tissues sections were then incubated with biotinylated anti-rabbit immunoglobulins (1/400, Dako, North Sydney, Australia), and streptavidin horseradish peroxidase (1/500, Dako, Australia), and then visualized using diaminobenzidine (DAB)/H2O2 substrates (Sigma Aldrich, Australia). Hematoxylin-stained TMA slides were Nanozoomer-scanned and quantified using ‘ImageJ’, coupled with ‘IHC profiler’ [27]. Further statistical analysis was performed using GraphPad Prism 6 v008 (GraphPad Software, La Jolla, CA, USA), where the mean, standard error of the mean, and significance (unpaired *t*-test) were calculated.

### 2.9. Online Database Analysis

The GENT2 database (data from the annotated Gene Expression Omnibus (U133Plus2)) was used to assess GSN, CALM1, and TXN mRNA levels in normal tissues, ovarian surface epithelium (n = 66), fallopian tube (n = 40), and HGSOC tissues (n = 807) [18]. The Kaplan–Meier Plotter (http://kmplot.com/, assessed on 7 December 2022) was used to investigate the relationship between the mRNA expression levels and the clinical outcome of the ovarian cancer cohort by combining mean expression of probes for GSN (GSN200696_s_at and 214040_s_at), CALM1 (209563_x_at, 200655_s_at, 209563_x_at, 211985_s_at, and 213688_at), and TXN (208864_s_at and 216609_at) which includes data from 15 datasets including Gene Expression Omnibus and The Cancer Genome Atlas [28]. Each gene was analyzed for both the progression-free survival (PFS) and overall survival (OS) against a database of 1029 and 1144 ovarian cancer patients (2022 version), respectively, using the following settings: patients were split by auto select best cut off, the histology was set to serous all stages, grades 2 and 3 were analyzed, all forms of chemotherapy were analyzed, and all biased arrays were excluded. 

## 3. Results

### 3.1. Identification of Protein Markers of Innate Chemoresistance by DIA LC-MS

Following protein extraction and trypsin digestion of the patient tissues and OVCAR-5 cell lines, samples were analyzed by DDA LC-MS/MS and data were analyzed using MaxQuant. The generated peptide sequence information allows the identification of proteins by matching them to a database. The DDA analysis of approximately 100 ng of tumor tissue samples resulted in the identification of 610 proteins (n = 6 chemo-responsive tissues, n = 6 non-responsive tissues) at a protein false-discovery rate (FDR) of 1% and with at least two unique peptides. Proteins identified are listed in the Appendix A. Analysis of approximately 400 ng from biological replicates of the parental and CBPR OVCAR-5 cells resulted in 1396 protein identifications at an FDR of 1%. 

For the analysis, spectral libraries were generated in the Skyline software including all DDA mass spectra and peptide identifications matched to a background proteome of human FASTA sequences downloaded from UniProt. All samples analyzed by DDA LC-MS/MS were then also analyzed by DIA LC-MS/MS. The acquired DIA mass spectra were then matched to peptide sequences contained in the spectral libraries and relative quantification was performed in Skyline. Following manual checking of the peptide matches in Skyline, GSN (fold change = 1.84, *p* value = 0.035), CALM1 (fold change = 2, *p* value = 0.029), and TXN (fold change = 2.47, *p* value = 0.0086) were found to be significantly increased in the cohort of patients who responded to chemotherapy compared to those who did not. Figure 1 shows the quantitative results of the DIA analysis of the 12 patient samples. The identification and quantification of GSN was based on three unique and unmodified peptides with 0 missed-cleavage and were as follows: HVVPNEVVVQR (aa178–188), EVQGFESATFIGYFK (aa148–162), and TPSAAYIWVGTGASEAEK (aa598–615). Similarly, CALM1 was identified and quantified based on 3 unique and unmodified peptides with 0 missed cleavage i.e., EAFSIFDK (aa15–22), ADQITEEQIAEFK (aa2–14), and EADIDGDGQVNYEEFVQMMTAK (aa62–83). TXN was identified and quantified based on 2 unique and unmodified peptides with 0 missed cleavage i.e., IEATINEIV (aa97–105) and TAFQEAIDAAGDK (aa9–21).

### 3.2. Verification of Protein Markers of Innate Chemoresistance by IHC

The differential expression of GSN, CALM1, and TXN was confirmed between the chemo-responsive (n = 21) and non-responsive (n = 8) patient cohorts by IHC using a TMA cohort. Quantitative analysis of immunostaining was performed using IHC Profiler-Image J [27]. The overall levels of positive staining were found to be significantly increased in the chemo-responsive patient group as compared to the non-responsive group for GSN (fold change = 1.39, *p* value = 0.0023), CALM1 (fold change = 3.4, *p* value < 0.0001), and TXN (fold change = 1.73, *p* value < 0.0001) as shown in Figure 2, consequently validating the results observed by the DIA LC-MS analysis. 

### 3.3. Marker Expression in Normal Tissues Compared to HGSOC and Survival Analysis

To explore if the expression of the three proteins of interest is altered in healthy epithelium of the ovary (OSE) or fallopian tubes (FT) compared to high-grade serous ovarian cancer (HGSOC), we analyzed mRNA expression data from the GENT2 database. The advantage of this dataset is that there are more than 900 patient samples included, however mRNA levels do not necessarily correlate with protein levels. CALM1 expression was significantly reduced in HGSOC compared to normal ovarian surface epithelium (OSE) but both GSN and TXN mRNA levels were increased in HGSOC compared to OSE (Figure 3). Both CALM1 and TXN mRNA levels were significantly decreased in HGSOC compared to fallopian tube (FT) epithelium (Figure 3). GSN expression was reduced in HGSOC compared to FT but did not reach statistical significance (Figure 3). These findings suggest that the reduced levels of GSN, CALM1, and TXN in the FT, which is thought to be the site of origin of the majority of HGSOC, may be also associated with the development of HGSOC [29,30].

### 3.4. Kaplan–Meier Outcome Analysis

The relationships with PFS and OS were analyzed for GSN, CALM1, and TXN at the mRNA level using the on-line tool Kaplan–Meier Plotter. OS of patients with elevated levels of GSN mRNA expression was significantly increased compared to those with low levels of expression (Figure 4A), but CALM1 expression was not associated with a notable change in OS or PFS (Figure 4B). PFS of patients with elevated levels of TXN mRNA expression was significantly higher than those with low levels of expression (Figure 4C). 

### 3.5. Marker Expression in Parental OVCAR-5 Cells Compared to Carboplatin-Resistant OVCAR-5 Cells

The relative protein expression of GSN, CALM1, and TXN was compared in biological replicates of parental OVCAR-5 cells and CBPR OVCAR-5 cells by DIA LC-MS in the same way as the patient tissue samples had been analyzed (Figure 5). Protein levels of GSN (fold change = 1.36, *p* value = 0.027), CALM1 (fold change = 3, *p* value = 0.0017), and TXN (fold change = 2.38, *p* value = 0.021) were all found to be significantly decreased in the CBPR OVCAR-5 cells compared to the parental cells. The differential expression of reduced protein levels in the resistant or non-responsive samples was the same for both patients and cell lines for all three proteins. A minimum of two proteotypic and unmodified peptides was used for the analysis of each protein. 

## 4. Discussion

Chemoresistance is one of the major challenges in treating ovarian cancer effectively. The high mortality rate from ovarian cancer has been attributed to late diagnosis with a five-year survival rate less than 30% compared to ~95% when the patients are diagnosed at the earliest stage, highlighting the need for early detection [1]. Multiple molecular features of chemoresistance have been investigated in detail, such as the role of oncogenes and transporter pumps or of the tumor-suppressor gene p53, however the mechanism remains poorly understood. There is an unmet need to firstly identify patients with innate chemoresistance and secondly to develop efficient second- and third-line chemotherapeutic treatments to overcome acquired chemoresistance.

GSN, an actin-binding protein, has previously been identified in cytosol and mitochondria and in blood plasma. It has been implicated in several cancers by inhibiting apoptosis and stabilizing mitochondria. Serum GSN levels are significantly reduced in patients with ovarian cancer compared with healthy patients [31] and have been shown to be a crucial factor in regulating chemoresistance in vitro [32]. High GSN expression has been shown to significantly correlate with longer OS and PFS in all-stage patients and subgroups with serous ovarian cancer [32]. GSN can be secreted and has been detected in serum and plasma as pGSN. High plasma levels correlate with poorer overall survival and relapse-free survival in patients with OVCA, which could reflect the overall tumor load or pGSN, promoting ovarian cancer survival through both autocrine and paracrine mechanisms. Plasma GSN was identified as an independent poor prognostic biomarker for PFS of ovarian cancer patients [33]. Although GSN has been described by others to be highly expressed in chemoresistant cancer cells, our data using patient tissue did not confirm this. There is the possibility that during the processing of the tissue sample, the secreted pGSN might be lost [34].

CALM1 is a multifunctional calcium-binding protein, which has been implicated in many signaling pathways regulating cancer. Previous studies have shown that CALM1 antagonists induce apoptosis by decreasing AKT activation and increasing caspase-8 expression or decreasing anti-apoptotic Bcl-2 and increasing pro-apoptotic Bax protein levels [35]. This is due to CALM1′s recruitment to the Fas death receptor-activated death-inducing signaling complex (DISC), where it binds with the survival signals, FLIP and Src to mediate death-receptor-controlled survival pathways [35]. In the current study, downregulated CALM1 mRNA showed a big variability across the patients. However, when the patients were grouped according to high and low expression of CALM1, no significant difference between the two groups regarding progression or overall survival was observed. It would be interesting to explore the correlation between protein and mRNA levels of CALM1 and the impact on cancer initiation and progression. To our knowledge this is the first study to show that CALM1 expression is reduced in cancer compared to normal tissues and further studies are required to understand its functional role in cancer. 

TXN is ubiquitously expressed and plays a role in many biological processes, such as redox signaling [36]. Elevated serum levels of TXN were observed in ovarian cancer compared to normal persons with non-cancer inflammatory disease [37], although, Criscuolo et al. showed reduced glutathione levels in HGSOC patients that did not respond to platinum-based therapy, resulting in the increased expression of thioredoxin reductase [38]. Several studies have shown that cisplatin can reduce activity of thioredoxin reductase, leading to the lower level of TXN in resistant cancer cells [39]. Similar to our observation, Huang et al. reported reduced TXN gene expression in cisplatin-resistant cells and reconstitution of TXN increased sensitivity to cisplatin [40]. These results demonstrate that further studies are required to confirm the TXN outcome.

## 5. Conclusions

In summary, we have performed a proteomics discovery approach using minimal FFPE patients’ samples to identify potential markers of innate chemoresistance, using a combined DDA and DIA label-free MS approach. Three key markers GSN, CALM1, and TXN were observed to be consistently higher in abundance in the chemo-responsive patient cohort, as compared to non-responsive patient cohort. Undertaking the same label-free MS strategy using CBPR OVCAR 5 cells compared with the parental cell line recapitulates the expression of these three proteins. This indicates that the chemoresistant cell lines might serve as a suitable model system for chemoresistance and could be used to determine the efficacy of new generations of chemotherapeutics, extending the OS of ovarian cancer patients in the future.

## Figures and Tables

**Figure 1 cancers-15-02172-f001:**
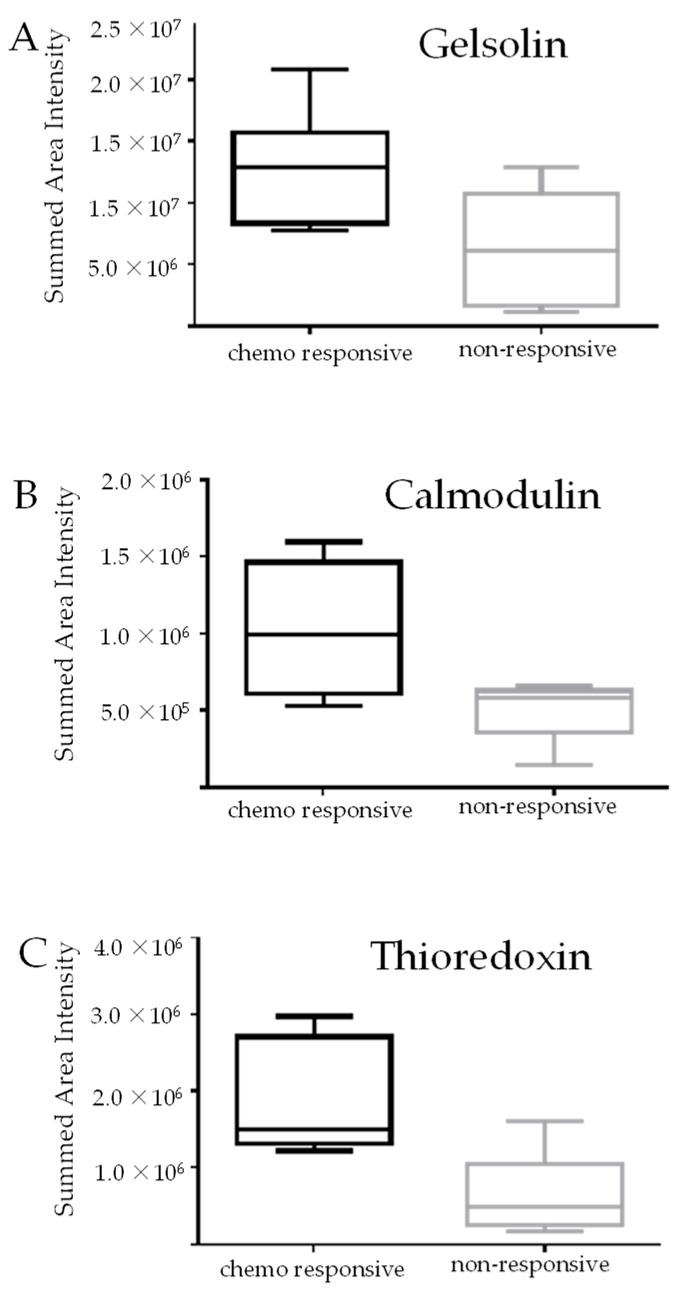
DIA analysis of HGSOC patients who responded to chemotherapy (n = 6) compared to those who did not (n = 6). Relative abundance of (**A**) gelsolin (fold change = 1.84, *p* value = 0.035), (**B**) calmodulin (fold change = 2, *p* value = 0.029), and (**C**) thioredoxin (fold change = 2.47, *p* value = 0.0086). Box and whisker plots were generated comparing the area intensities of each protein in the chemo- responsive and non-responsive patient cohorts. Significance was calculated using un-paired *t*-tests using the GraphPad Prism. The error bars indicate the standard deviation.

**Figure 2 cancers-15-02172-f002:**
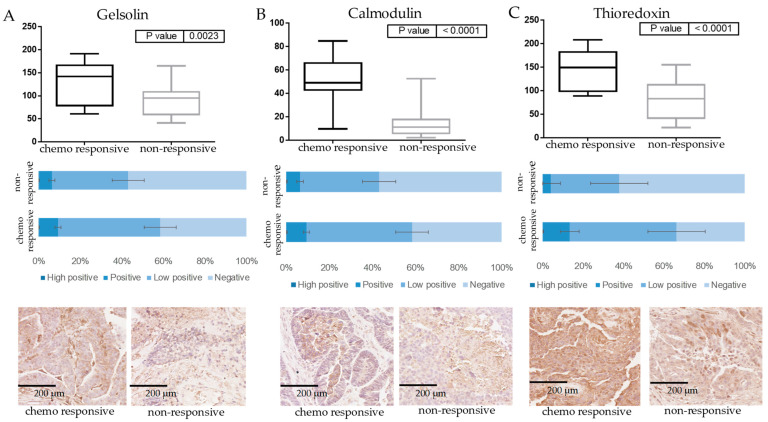
Immunohistochemical staining of gelsolin, calmodulin, and thioredoxin. IHC staining comparing ovarian tumors from patients who did not respond (n = 8) to chemotherapy to those who did (n = 21) for gelsolin, *p* value = 0.0023 (**A**), calmodulin, *p* value < 0.0001 (**B**), and thioredoxin, *p* value < 0.0001 (**C**). The graphs show the level of high-positive, positive, low-positive, and negative staining for the chemo-responsive patients compared to the non-responsive patient cohort. Representative image of stained tissue at 20× magnification from the responsive and non-responsive patients is shown at the bottom of each graph. The level of positive staining across the patients was summed and significance was calculated using un-paired *t*-test using GraphPad prism. Quantitative analysis was performed using IHC profiler-Image J. For each tissue section, three representative photo-micrographic images at 40× magnification were used.

**Figure 3 cancers-15-02172-f003:**
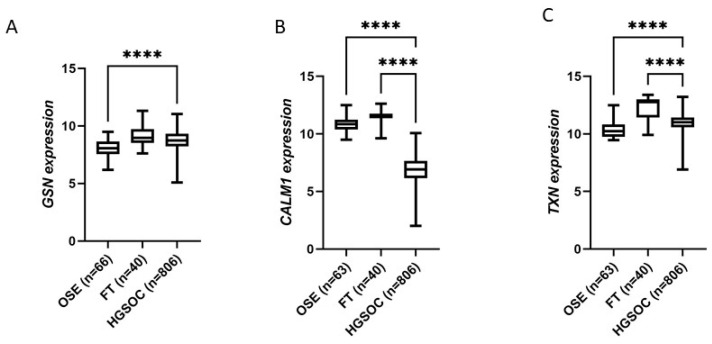
Marker expression in normal tissues compared to HGSOC. GSN (**A**), CALM1 (**B**), and TXN (**C**) mRNA expression data obtained from the GENT2 database including ovarian surface epithelium (OSE) (n = 66), fallopian tube (FT) epithelium (n = 40), and high-grade serous ovarian cancer HGSOC (n = 807). **** *p* < 0.0001, Kruskal–Wallis with Dunn’s multiple comparison test.

**Figure 4 cancers-15-02172-f004:**
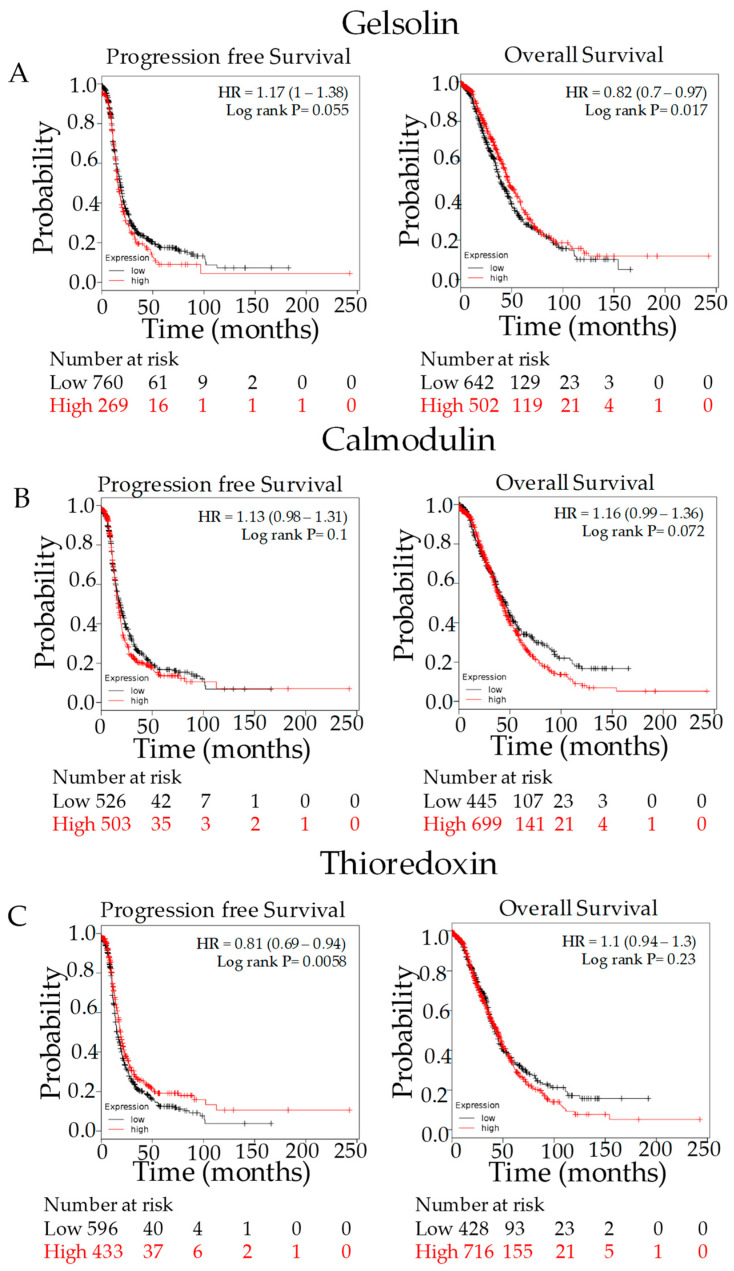
Kaplan–Meier plot using online TCGA data. Kaplan–Meier analysis revealed that elevated levels of GSN (**A**) and TXN (**C**) mRNA expression correlated with better five-year OS and PFS rates, respectively. CALM1 (**B**) expression was not associated with OS or PFS. Analysis was performed on data from 1648 ovarian cancer patients downloaded from Gene Expression Omnibus and The Cancer Genome Atlas using the Kaplan–Meier Plotter (http://kmplot.com/ assessed on 12 January 2022). Higher expression is shown in red while low is shown in black.

**Figure 5 cancers-15-02172-f005:**
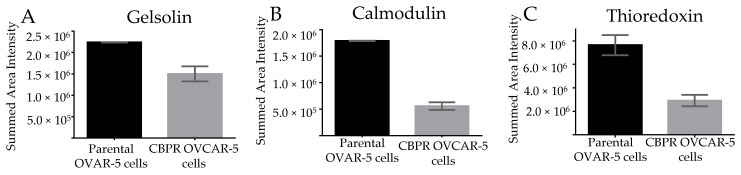
DIA analysis of parental OVCAR-5 cells compared to CBPR OVCAR-5 cells. Relative quantification of gelsolin *p* value = 0.027 (**A**), calmodulin *p* value = 0.0017 (**B**), and thioredoxin *p* value = 0.021 (**C**) in biological replicates of parental OVCAR-5 cells compared to CBPR OVCAR-5 cells. Graphs show the standard error of the mean and significance was calculated using an unpaired *t*-test.

**Table 1 cancers-15-02172-t001:** Clinicopathological information for the HGSOC patients used in the study.

Patient	First Treatment	Grade	Category	Age at Diagnosis (Years)
1	Carboplatin/paclitaxel	3	Complete response	63
2	Carboplatin	3	Complete response	69
3	Carboplatin/paclitaxel	3	Complete response	61
4	Carboplatin/paclitaxel	3	Complete response	64
5	Carboplatin	3	Complete response	60
6	Carboplatin/paclitaxel	3	Complete response	59
7	Carboplatin/paclitaxel	3	Non-response/incomplete response	78
8	Carboplatin/paclitaxel	3	Non-response/incomplete response	66
9	Carboplatin/paclitaxel	3	Non-response/incomplete response	61
10	Carboplatin/paclitaxel	3	Non-response/incomplete response	44
11	Carboplatin/paclitaxel	3	Non-response/incomplete response	75
12	Carboplatin	3	Non-response/incomplete response	78

## Data Availability

The mass spectrometry proteomics data have been deposited to the ProteomeXchange Consortium via the PRIDE [41] partner repository with the dataset identifier PXD033785 and 10.6019/PXD033785.

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
