# Peer review of "Label-Free Quantification Mass Spectrometry Identifies Protein Markers of Chemotherapy Response in High-Grade Serous Ovarian Cancer"

_cancers, 2023, doi:10.3390/cancers15072172_

Round 1

Reviewer 1 Report

The authors of “Label Free Quantification Mass Spectrometry Identifies Protein Markers of Chemotherapy Response in High Grade Serous Ovarian Cancer” aim to find biomarkers of innate chemoresistance. They do this using a relative quantitation MS approach which is then validated by an IHC approach. The authors seek to understand better the implications of these markers by using online database analysis. Additionally, they introduce a cell model capable of replicating the biomarkers’ expression.

To my knowledge, this article contains some novelty aspects which are relevant for the field and the special issue which is inserted in. In general, the article is well structured. That said it could be further improved/enriched and clarified.

Please answer or correct the following aspects:

·       Line 45: I believe the sentence should be rephrased since we are already in 2023.

·       Line 85: It could be interesting to comment (in the results and/or discussion) if you were able to identify or not any of the markers mentioned here.

·       Line 88: Explain the HGSOC abbreviation.

·       Line 95: Explain the FIGO abbreviation.

·       Line 104: Since you showed that the GSN reduction was not significant, I would rephrase this.

·       Line 105: Explain the FT abbreviation and add the reference at “(add ref.)”.

·       Line 120: Correct “written inform consent”, it should be “informed”. Correct wherever necessary.

·       Line 130: You could summarize some information from the reference that is relevant to this article or add a supplementary table. Please check again if ref. 18 has the same information mentioned in the present article (treatment and age).

·       Line 142: Give details on the microscope model used.

·       Line 148: Add “total” before “protein concentration”.

·       Line 157: Add “half maximal” before “inhibitory concentration (IC50)”.

·       Line 163: Add "(results not shown)" referring to the IC50 curves.

·       Line 168: Add ™ after “EZQ”.

·       Line 175: Can you elaborate on the reason why you use 20 °C for the protein reduction with DTT? I’m curious to know about the efficiency of this process, since the temperature commonly used is around 56 °C. Or was it a typo?

·       In the 2.5 section I suggest the indication of the mass spectrometer’s source settings as well.

·       Line 211: Add peptide/precursor before “mass error tolerance”, so that it’s more clear which type of ion you are referring to. Additionally, I believe “carbamidomethyl” should be written “carbamidomethylation” instead.

·       Line 223: I cannot check if the PRIDE submission has these Maxquant msms files, but I believe it would be important to add those. If not, you could add what type of important information they contain, which is used for the present analysis.

·       Line 229: In the text, “precursor charge states” and “ion charges” seem redundant. I know the software might have these names, but I believe it would be more clear (for those who are not familiar with the software) to have “product ion charges” in the second case.

·       Line 231: You mention a 10 000 resolution, but I imagine the resolution on your TOF would be set higher and I believe the default of Skyline is 30 000. Can you comment on why you chose to reduce the resolution setting here? Additionally, you used only a minimum of 2 proteotypic peptides. Since a minimum of 3 is commonly used, can you also comment on why you had to reduce?

·       Line 235: Add "(GraphPad Software, La Jolla, CA, USA)" reference, which is used in line 259.

·       Line 260: Here you mention “paired t-test”, but in Figure 2 you mention “unpaired”. Is “paired” a typo? Additionally, can you also elaborate on the negative and positive controls used?

·       Line 262: On section 2.9 it seems more natural to me if the GENT2 description comes before the Kaplan-Meier description, just for the sake of being in the same order as in the Results.

·       Line 284: In the Supplementary table 1 I could count 611 protein groups, but in the article, you mention 607. Is this a typo or can you comment on this discrepancy?

·       Line 287: Here is the first time that biological replicates are mentioned. I believe it’s important to also mention them in the Methods section, along with details of the experimental design, including how many were used.

·       Line 294: It would be useful to know how the “manual checking” was performed or to elaborate on what you mean. I suggest you mention the criteria, etc; or show the XICs of the peptides/transitions in question, taken from Skyline. You could even remove the peptide sequences from the text and put them in the figure only.

·       Line 300: Correct “mis-cleavage”, it should be “missed cleavage”. Correct wherever necessary.

·       Line 314: Add “innate” before “chemoresistance”, for the sake of clarity and continuity on the previous section title.

·       Line 334: I believe the survival analysis part is addressed in the next section and not in this one, as the title says. Additionally, I believe the beginning of this section could benefit from a small introduction of why this part was performed. As it is, it seems somehow blunt going from Protein analysis to mRNA database analysis.

·       Line 340: Be careful with including the loss of GSN, as it was not significant. I would rephrase this.

·       Lines 341-347: Correct the references, which were not properly added.

·       Line 357: I believe the observation related to CALM1 was not discussed in the Discussion section. I think it would be interesting to add.

·       Line 399-405 and line 415-417: Further discussion on how this relates to your results would make it clearer and more interesting.

·       Lines 498-499: There is a repeated reference.

·       Since you mention other markers in the introduction (such as p53, etc.), I believe the article could benefit from a comparison and discussion on the specificity and sensitivity of the new markers (GSN, etc.).

.        Add the following abbreviations to the appropriate section: FIGO, FT, OS, PFS.

Author Response

Manuscript ID: cancers-2228604 entitled “Label Free Quantification Mass Spectrometry Identifies Protein Markers of Chemotherapy Response in High Grade Serous Ovarian Cancer”
We like to thank the reviewers for their efforts and helpful comments. We have carefully considered all comments as indicated in detail below and believe that the revised version meets the journal publication requirements.
Reviewers’ comments in black, response in cursive and blue
Author’s response to the reviewers’ comments:
Reviewer 1
The authors of “Label Free Quantification Mass Spectrometry Identifies Protein Markers of Chemotherapy Response in High Grade Serous Ovarian Cancer” aim to find biomarkers of innate chemoresistance. They do this using a relative quantitation MS approach which is then validated by an IHC approach. The authors seek to understand better the implications of these markers by using online database analysis. Additionally, they introduce a cell model capable of replicating the biomarkers’ expression.
To my knowledge, this article contains some novelty aspects which are relevant for the field and the special issue which is inserted in. In general, the article is well structured. That said it could be further improved/enriched and clarified.
Please answer or correct the following aspects:
Line 45: I believe the sentence should be rephrased since we are already in 2023.
The Cancer Australia website was last updated on 1st September 2022 so that’s the most recent data.
No changes have been made.
Line 85: It could be interesting to comment (in the results and/or discussion) if you were able to identify or not any of the markers mentioned here.
However, none of the markers mentioned here have been identified in our proteomics study, indicating that they are low abundant.
This sentence has been added to the manuscript.
Line 88: Explain the HGSOC abbreviation.
Edited as advised
Line 95: Explain the FIGO abbreviation.
Edited as advised
Line 104: Since you showed that the GSN reduction was not significant, I would rephrase this.
Edited as requested
Line 105: Explain the FT abbreviation and add the reference at “(add ref.)”.
Edited as advised
Line 120: Correct “written inform consent”, it should be “informed”. Correct wherever necessary.
Edited as advised
Line 130: You could summarize some information from the reference that is relevant to this article or add a supplementary table. Please check again if ref. 18 has the same information mentioned in the present article (treatment and age)
As per the reviewer’s recommendation, supplementary table 1 is added containing all the relevant patient’s information needed for the paper
Line 142: Give details on the microscope model used.
Edited as requested
Line 148: Add “total” before “protein concentration”
Edited as advised
Line 157: Add “half maximal” before “inhibitory concentration (IC50)”.
Edited as advised
Line 163: Add "(results not shown)" referring to the IC50 curves.
Edited as advised
Line 168: Add ™ after “EZQ”.
Edited as advised
Line 175: Can you elaborate on the reason why you use 20 °C for the protein reduction with DTT? I’m curious to know about the efficiency of this process, since the temperature commonly used is around 56 °C. Or was it a typo?
The temperature was kept low to reduce the rate at which urea decomposes. Heating the urea containing buffers at high temperature, can increase the risk of carbamylation that eventually affects the enzymatic digestion of the proteins and results in poor protein identification.
No changes have been made.
In the 2.5 section I suggest the indication of the mass spectrometer’s source settings as well.
Edited as requested
Line 211: Add peptide/precursor before “mass error tolerance”, so that it’s more clear which type of ion you are referring to. Additionally, I believe “carbamidomethyl” should be written “carbamidomethylation” instead.
Edited as advised
Line 223: I cannot check if the PRIDE submission has these Maxquant msms files, but I believe it would be important to add those. If not, you could add what type of important information they contain, which is used for the present analysis.
All relevant information including the raw (.baf) files, mgf files and corresponding resultant files in mzid and csv format are uploaded on proteome exchange.
No changes have been made.
Line 229: In the text, “precursor charge states” and “ion charges” seem redundant. I know the software might have these names, but I believe it would be more clear (for those who are not familiar with the software) to have “product ion charges” in the second case.
Edited as advised
Line 231: You mention a 10 000 resolution, but I imagine the resolution on your TOF would be set higher and I believe the default of Skyline is 30 000. Can you comment on why you chose to reduce the resolution setting here? Additionally, you used only a minimum of 2 proteotypic peptides. Since a minimum of 3 is commonly used, can you also comment on why you had to reduce?
The resolution was reduced for MS2 as described in the manuscript. We have replaced proteotypic peptides with unique peptides in the manuscript.
Line 235: Add "(GraphPad Software, La Jolla, CA, USA)" reference, which is used in line 259.
Edited as advised
Line 260: Here you mention “paired t-test”, but in Figure 2 you mention “unpaired”. Is “paired” a typo? Additionally, can you also elaborate on the negative and positive controls used?
Thank you for pointing out our mistake, unpaired t-test was used for the analysis. The manuscript is edited accordingly.
We used two standard negative reagent (NRC) controls; we relaced the primary antibody with non-specific IgG (here 5% goat serum) and in the second NRC omitted the primary antibody. The performance of the secondary antibody had been tested before.
Line 262: On section 2.9 it seems more natural to me if the GENT2 description comes before the Kaplan-Meier description, just for the sake of being in the same order as in the Results.
Edited as advised
Line 284: In the Supplementary table 1 I could count 611 protein groups, but in the article, you mention 607. Is this a typo or can you comment on this discrepancy?
Thank you for pointing out our mistake, it was a typo and edited accordingly.
Line 287: Here is the first time that biological replicates are mentioned. I believe it’s important to also mention them in the Methods section, along with details of the experimental design, including how many were used.
In total twelve tissue sections were used for the LC-MSMS study including 6 patients that responds to chemotherapy while 6 patients that has not responded. As advised, the manuscript is edited.
Line 294: It would be useful to know how the “manual checking” was performed or to elaborate on what you mean. I suggest you mention the criteria, etc; or show the XICs of the peptides/transitions in question, taken from Skyline. You could even remove the peptide sequences from the text and put them in the figure only.
Thanks for the suggestion, as advised the sentence was rewritten in the manuscript as “The DIA data was analysed in Skyline software, where the transition and the retention time for each peptide was checked manually. The quantification was performed using the summed area intensity of each peptide in each sample”
Line 300: Correct “mis-cleavage”, it should be “missed cleavage”. Correct wherever necessary.
Edited as advised.
Line 314: Add “innate” before “chemoresistance”, for the sake of clarity and continuity on the previous section title.
Edited as advised.
Line 334: I believe the survival analysis part is addressed in the next section and not in this one, as the title says. Additionally, I believe the beginning of this section could benefit from a small introduction of why this part was performed. As it is, it seems somehow blunt going from Protein analysis to mRNA database analysis.
As advised, following sentences has been added in the revised manuscript “To explore if the expression of the three proteins of interest is altered in healthy epithelium of the ovary (OSE) or fallopian tubes (FT) compared to high-grade serous ovarian cancer (HGSOC), we analysed mRNA expression data from the GENT2 database. The advantage of this dataset is that there
are more than 900 patient samples included, however mRNA levels do not necessarily correlate with protein levels”.
Line 340: Be careful with including the loss of GSN, as it was not significant. I would rephrase this.
The sentence has been rephrased from “the loss of GSN, CALM1 and TXN” to “reduced levels of CALM1 and TXN”
Lines 341-347: Correct the references, which were not properly added.
Edited as advised
Line 357: I believe the observation related to CALM1 was not discussed in the Discussion section. I think it would be interesting to add.
Following sentences has been added in the revised manuscript “Down regulated CALM1 mRNA showed a big variability across the patients. However, when the patients were grouped according to high and low expression of CALM1, no significant difference between the two groups regarding progression free or overall survival was observed. It would be interesting to explore the correlation between protein and mRNA levels of CALM1 and the impact on cancer initiation and progression.”
Line 399-405 and line 415-417: Further discussion on how this relates to your results would make it clearer and more interesting.
Following sentences has been added in the revised manuscript “: GSN, an actin-binding protein, has previously been identified in the cytosol. It has been implicated in several cancers by inhibiting apoptosis and stabilizing mitochondria. GSN can get secreted and has been detected in serum and plasma as pGSN. High plasma levels correlate with poorer overall survival and relapse-free survival in patients with OVCA, which could reflect the overall tumour load or of pGSN promoting ovarian cancer survival through both autocrine and paracrine mechanisms. Plasma GSN was identified as an independent poor prognostic biomarker for PFS of ovarian cancer patients https://doi.org/10.1016/j.mcpro.2023.100502. Although GSN has been described by others to be highly expressed in chemoresistant cancer cells, our data using patient tissue did not confirm this. There is the possibility that during the processing of the tissue sample, the secreted pGSN might be lost. https://www.nature.com/articles/s41388-019-1087-9
Lines 498-499: There is a repeated reference.
Edited as advised
Since you mention other markers in the introduction (such as p53, etc.), I believe the article could benefit from a comparison and discussion on the specificity and sensitivity of the new markers (GSN, etc.).
As per the reviewer's recommendation, discussion has been rewritten
Add the following abbreviations to the appropriate section: FIGO, FT, OS, PFS.
Edited as advised

Reviewer 2 Report

The authors identified 3 proteins that were increased in responders versus non-responders of HGSOC (GSN, CALM1 and TXN) which correlated with TMA and the ovarian cell line OVCAR5 versus its carboplatin resistant variant.   Kaplan-Meier analysis showed GSN to correspond to overall survival and TXN to PFS.  The result were presented well but the discussion could have better placed their results in comparison to current research findings in the area.  A quick trawl revealed papers on GSN (Pierredon, 2017; Kim 2023, Punzén-Jimenés 2022; and Picktel 2018), Calmodulin related proteins (Previs, 2019; Shepherd, 2022; Gocher, 2017) and TXN (Raninga 2022).

The authors should also compare the results obtained by Criscuolo (2022) who found increased TrxR activity in resistant HGSOC cells.

The authors should define HGSOC when first mentioned

Line 16..to add reference??

line 341...full references in text.. needs to be fixed

references 32 and 33 are the same citation

Author Response

Manuscript ID: cancers-2228604 entitled “Label Free Quantification Mass Spectrometry Identifies Protein Markers of Chemotherapy Response in High Grade Serous Ovarian Cancer”
We like to thank the reviewers for their efforts and helpful comments. We have carefully considered all comments as indicated in detail below and believe that the revised version meets the journal publication requirements.
Reviewers’ comments in black, response in cursive and blue
Author’s response to the reviewers’ comments:
Reviewer 2
The authors identified 3 proteins that were increased in responders versus non-responders of HGSOC (GSN, CALM1 and TXN) which correlated with TMA and the ovarian cell line OVCAR5 versus its carboplatin resistant variant. Kaplan-Meier analysis showed GSN to correspond to overall survival and TXN to PFS. The result were presented well but the discussion could have better placed their results in comparison to current research findings in the area. A quick trawl revealed papers on GSN (Pierredon, 2017; Kim 2023, Punzén-Jimenés 2022; and Picktel 2018), Calmodulin related proteins (Previs, 2019; Shepherd, 2022; Gocher, 2017) and TXN (Raninga 2022 https://doi.org/10.18632/oncotarget.3795).
The authors should also compare the results obtained by Criscuolo (2022) who found increased TrxR activity in resistant HGSOC cells. https://www.ncbi.nlm.nih.gov/pmc/articles/PMC9404763/
Thanks for the suggestions, as per the reviewer's recommendation discussion has been rewritten
The authors should define HGSOC when first mentioned
Edited as advised
Line 16..to add reference??
Edited as advised
line 341...full references in text needs to be fixed
Edited as advised
references 32 and 33 are the same citation
Edited as advised
